# Study of Dispersed Repeats in the *Cyanidioschyzon merolae* Genome

**DOI:** 10.3390/ijms25084441

**Published:** 2024-04-18

**Authors:** Valentina Rudenko, Eugene Korotkov

**Affiliations:** Institute of Bioengineering, Research Center of Biotechnology of the Russian Academy of Sciences, Moscow 119071, Russia; bioinf@yandex.ru

**Keywords:** dispersed repeats, transposable elements, repeated DNA, *Cyanidioschyzon merolae*, alignment algorithm, position weight matrix

## Abstract

In this study, we applied the iterative procedure (IP) method to search for families of highly diverged dispersed repeats in the genome of *Cyanidioschyzon merolae*, which contains over 16 million bases. The algorithm included the construction of position weight matrices (PWMs) for repeat families and the identification of more dispersed repeats based on the PWMs using dynamic programming. The results showed that the *C. merolae* genome contained 20 repeat families comprising a total of 33,938 dispersed repeats, which is significantly more than has been previously found using other methods. The repeats varied in length from 108 to 600 bp (522.54 bp in average) and occupied more than 72% of the *C. merolae* genome, whereas previously identified repeats, including tandem repeats, have been shown to constitute only about 28%. The high genomic content of dispersed repeats and their location in the coding regions suggest a significant role in the regulation of the functional activity of the genome.

## 1. Introduction

Recent advances in the development of sequencing technologies have led to the accumulation of large data on whole genome sequences of various biological species, which need to be annotated for further application in biotechnology, medicine, and scientific research [1].

It is now established that a large portion of the eukaryotic genome is occupied by repeated sequences; thus, they constitute 85% in some cereals [2], 81% in peppers [3], and up to 90% in some fish species [4]. These repeated sequences are usually dispersed throughout the genome and are typically mapped to heterochromatic, gene-poor, or intergenic regions, where the repeat density can sometimes reach 100% [5]. Initially, the repeated sequences were considered to play a minor role in the genome, but accumulating data indicate that repeats represent a source of genetic variations, providing better adaptability of the organism to the environment [6].

In the eukaryotic genome, a considerable part of dispersed repeats is represented by transposable elements (TEs). Most TEs encode proteins that mediate their autonomous transmission and are classified according to the mechanisms of transposition and chromosomal integration. The first TE class is represented by retrotransposons [7], which spread through the “copy-and-paste” mechanism, including the formation of an RNA intermediate; in turn, they are subdivided into those with and without long terminal repeats (LTRs and non-LTRs, respectively). The second TE class comprises DNA transposons that do not use reverse transcription and move either through the “cut-and-paste” mechanism or “peel-and-paste” replication with the participation of a circular DNA intermediate [8,9]. However, this classification may change as data on new TE types are constantly emerging [10].

Since the movement and accumulation of dispersed repeats represent a major force shaping the genomes of almost all organisms, their analysis is important for understanding the impact of TEs on genome evolution. It is believed that repeated sequences are the major contributors to genomic instability due to possible recombination between similar sequences, which can lead to chromosomal rearrangements [11]. Thus, genes located closer to TEs have a higher mutation rate and are usually responsible for phenotypic variability within a species [12]. It has also been suggested that TEs could participate in the restructuring of gene regulatory networks [13].

Currently, there are two groups of methods for identifying dispersed repeats in genomes [14]. The first compares the genome with already known repeat sequences from specialized databases such as Dfam [15] or Repbase [16]. Dfam represents a collection of multiple sequence alignments for the members of a specific repeat family, which can be used to generate hidden Markov models; the database contains 3,437,876 repeat families (including 18,730 curated) from 2306 species. Repbase includes repeats from more than 100 organisms and is used in genome sequencing projects as a reference collection for masking repetitive DNA with software tools such as RepeatMasker [17] (https://repeatmasker.org/ (accessed on 16 April 2024)) or CENSOR (https://www.girinst.org/downloads/software/censor/ (accessed on 16 April 2024)) [18].

In contrast, the second group of methods finds dispersed repeats de novo without using any prior information about repeat composition and structure and considers only the genomic sequence in question. These methods are based on two approaches: k-mer calculation and self-comparison of the analyzed sequence. In the first, DNA regions in which the concentration of different k-mers shows statistically significant deviation from the random level are recognized as locations of potential repeats. The existing algorithms for finding dispersed repeats based on k-mers can include expansion of the region with non-random k-mer distribution [19,20,21], training a classifier on areas with high k-mer frequencies [22], k-mers assembly [23,24], or grouping them into clouds [25].

The self-comparison methods apply fast sequence similarity tools such as BLAST (https://blast.ncbi.nlm.nih.gov/Blast.cgi (accessed on 16 April 2024)) and then cluster the results, allowing for the construction of a repeat consensus and assignment of a particular repeat to a specific family, which is a significant advantage over the k-mer calculation. However, building a consensus is often a challenge. Thus, it should be taken into account that individual repeat families may include other (shorter) repeated elements, and it is necessary to correctly determine the complex structure of a particular repeat. Furthermore, the repeats of one family can be fragmented by the insertion of repeats from other families; in this case, we deal with the fragments of repeats from different families, which can also complicate the construction of a family consensus. Since there is no universal method for determining all types of dispersed repeats, pipelines that combine several tools are frequently used [26,27,28,29].

As a rule, dispersed repeats are located in non-coding DNA regions and accumulate a large number of mutations, which impedes their identification. The majority of the existing bioinformatics methods, regardless of whether they use specialized libraries or search for repeats de novo, can find dispersed repeats if the average number of substitutions per nucleotide between two family repeats (*x*) is ≤1.0 and fail to do so at *x* > 1.0; thus, they are not effective if the members of a family have accumulated a large number of base substitutions [30]. Previously, we have developed a method based on the iterative procedure (IP), which allows for finding repeat families for *x* ≤ 1.5 and applied this method to identify dispersed repeats in the *Escherichia coli* genome [30]. As a result, families of highly divergent dispersed repeats with *x* > 1.0 that could not be previously detected by other methods have been identified. The found repeats cover approximately 50% of the *E. coli* genome and mostly represent specific motifs of the bacterial genes. It can be hypothesized that in bacteria, dispersed repeats may take part in the compaction of DNA into a nucleoid [30].

The results obtained in search of highly diverged dispersed repeats in the bacterial genome prompted us to use the IP method to look for new repeat families in the eukaryotic genome. Since complete analysis of a genomic sequence by the IP method requires considerable computing time, for this study, we chose an organism with a relatively short genome. A unicellular red alga *Cyanidioschyzon merolae* was analyzed using the IP method as an approach to obtain position weight matrices (PWMs) for families of dispersed repeats. The generated PWMs were then applied to comprehensively identify dispersed repeats by dynamic programming after considering the correlation between neighboring nucleotides. The results revealed that the *C. merolae* genome contained 20 repeat families comprising, in total, 33,938 dispersed repeats, which is significantly more than has been previously found using other methods. The repeats occupied over 72% of the *C. merolae* genome, indicating its low complexity.

*C. merolae* is a eukaryotic unicellular red alga from phylum Rhodophyta, which lives in hot springs with pH < 2, a temperature of 45 °C, and high sulfur concentration. *C. merolae* cells are about 2 μm in size and contain one chloroplast and one mitochondria. The organism was chosen for this study because among non-symbiotic eukaryotes, it has the simplest nuclear genome, which has been fully characterized, including repeat sequences; the sizes of its nuclear, chloroplast, and mitochondrial DNA are 16,546,747 bp, 149,987 bp, and 32,211 bp, respectively (https://plants.ensembl.org/Cyanidioschyzon_merolae/Info/Annotation/#assembly (accessed on 16 April 2024)). The *C. merolae* genome consists of 20 chromosomes carrying rather few genes—5331, of which 4984 are annotated; introns are found in only 26 genes, all but one of which have just one intron [31]. *C. merolae* has been shown to contain the smallest known histone gene cluster, a unique telomeric repeat at all chromosome ends, and very few transposons [32].

Bioinformatic studies have revealed that repetitive sequences in the *C. merolae* genome (annotated in http://plants.ensembl.org (accessed on 16 April 2024)) constitute slightly more than 28%. The repeats have been identified using Ensembl Genomes repeat feature pipelines, which include the following software tools: DUST (https://github.com/lh3/sdust (accessed on 16 April 2024)) [33], Tandem Repeat Finder (TRF) (https://github.com/Benson-Genomics-Lab/TRF/releases/tag/v4.09.1 (accessed on 16 April 2024)) [34], RepeatMasker (https://repeatmasker.org/ (accessed on 16 April 2024)) [17], Repeat Detector (RED) (https://github.com/DionLab/RepeatDetector (accessed on 16 April 2024)) [22], and Ensembl/plant-scripts (https://github.com/Ensembl/plant-scripts (accessed on 16 April 2024)) [35]. The capabilities of these programs are briefly summarized below.

1. DUST masks low-complexity sequences and improves the quality of alignment in search for similarity; it is recommended to be used prior to the other algorithms, as the removal of low-complexity sequences from consideration reduces the calculation time.

2. TRF is the standard most commonly used software for finding tandem repeats, which may have nucleotide insertions and deletions (indels) relative to the consensus. However, TRF can only find repeats with a high degree of similarity (>50%).

3. RepeatMasker searches for dispersed repeats and low-complexity regions in DNA sequences using the Dfam database to build hidden Markov models (HMM) of repeat profiles; it can also extract consensus sequences from other databases. In the annotation of the *C. merolae* genome, the MIPS Repeats database and the plant-specific nrTEplants library with curated repeats from REdat, RepetDB, and TREP have also been used [36].

4. RED uses machine learning techniques for the de novo search of simple repeats or transposons after training on the analyzed genome. It calculates k-mer frequencies and identifies regions with many frequently repeated k-mers and those in which repeats are not expected and uses them to build an HMM model.

5. Ensembl/plant-scripts apply a hybrid approach based on counting k-mer frequencies and comparing them with curated repeat libraries. In terms of the number of identified repeats, Ensembl/plant-scripts is similar to RED, but operates faster, which is an advantage when a large number of genomes are analyzed.

## 2. Results

### 2.1. Search for Repeats in the C. merolae Genome

The algorithm to search for dispersed repeats in the *C. merolae* genome is described in detail in the Materials and Methods (Section 4.1). First, the PWM for each repeat family was determined using the IP method, and new families were considered if the number of repeats in a family *N_max_* (Section 4.1.4) exceeded 300. This threshold was chosen because the average number of elements in a family for a randomly shuffled sequence is 122 (*σ* = 12), and a family with *N_max_* ≥ 300 elements corresponds to deviation from a random family by *σ* > 10.0, which indicates that the size of the generated family is not random. Under these conditions, we were able to find 20 repeat families and create the PWM for each using the IP method, which was designated as *Mt_max_* (Materials and Methods, Section 4.1.4).

Then, we searched for dispersed repeats of each family in the *C. merolae* genome using the generated PWMs designated as *Mt_max_*. In parallel, a similar search was conducted in a randomly shuffled *C. merolae* genome designated *Rand*. We determined the number of repeats in each family of the *C. merolae* and *Rand* genomes and calculated the False Discovery Rate (FDR) as: FDR = FP/(FP + TP) (where FP and TP are false and true positives, respectively) in order to determine a threshold value of *Z*_0_ with FDR ≤ 4.0%. The results showed that all families except 3, 12, 14, 19 (*Z*_0_ = 5.0) and 3, 12, 14, 19 (*Z*_0_ = 5.5) had FDR ≤ 4.0%. In further analysis, we only considered dispersed repeats with *Z ≥ Z*_0_. Table 1 shows the number of repeats detected in each family of the *C. merolae* and *Rand* genomes, and Table 2 shows total repeat numbers in the families of each *C. merolae* chromosome on the forward and reverse DNA strands. In total, we identified 33,938 repeats, which occupied 12,019,586 bp or 72.64% of the *C. merolae* genome. The data illustrating the distribution of dispersed repeats from different families across chromosomes of *C. merolae* are presented as circos plots in Appendix A (*circos_plot_<n>.pdf*), where *n* is the family number from 1 to 20.

All found repeats are presented in Appendix A; the repeats detected on the forward (cyanidio_repeats_dir.csv) and reverse (cyanidio_repeats_inv.csv) DNA strands are shown separately.

We also constructed a histogram showing the length distribution of all detected repeats (Figure 1). The minimum, maximum, and average lengths of the repeats identified by the IP method were 108, 600, and 522 bp, respectively. Interestingly, the repeat length distribution had two local peaks of 360 and 560 bp.

As the location of the genes in the *C. merolae* genome is known, we analyzed the intersections between the genes and the identified repeats. An intersection was considered if a repeat overlapped with a gene or if a gene overlapped with a repeat by more than 50% of the respective length. The results indicated that 14,187 of the repeats were located in the genes and 4288 *C. merolae* genes, i.e., more than 86% of those annotated contained repeats.

### 2.2. Calculation of Consensus Sequences for the Repeat Families

Each found repeat family was characterized by its own *Mt_max_*, which showed the weight of each base at each position of the repeat. If *Mt_max_* is greater than zero, then there is more probability for a specific base to be at a given position than is expected for a random sequence. To demonstrate this clearly, we constructed a consensus sequence for each repeat family and calculated these consensuses in the numerical, symbolic, and Weblogo formats.

To build the consensus, we determined matrix *M*(4600) for each family using pairwise alignment of each repeat with *Mt_max_* columns of this family. For example, a fragment of the alignment from columns 21 to 33 might look as follows:
(1)$$\begin{array}{c} \mathrm{21}\;\mathrm{22}\;\mathrm{23}\;\mathrm{24}\;\mathrm{25}\;\mathrm{26}\;*\;\mathrm{27}\;\mathrm{28}\;\mathrm{29}\;\mathrm{30}\;\mathrm{31}\;\mathrm{32}\;\mathrm{33}\; \end{array}$$
$$\begin{array}{c} a\;\;t\;\;c\;\;c\;\;g\;\;g\;\;t\;\;a\;\;c\;\;c\;\;*\;\;c\;\;t\;\;g \end{array}$$

Here, the top row is the sequence of *Mt_max_* columns denoted as *a*(*i*) (*i* = 1, 2, …, 600), and the bottom row is the nucleotide sequence of the found repeat denoted as *s*(*i*); asterisks indicate deletion at a given position. Matrix *M* was calculated for all alignments of the repeat family as: *M*((*s*(*k*),*a*(*k*)) = *M*((*s*(*k*),*a*(*k*)) + 1 for all *k* from 1 to *L*. If *s*(*k*) = 0 or *a*(*k*) = 0 (nucleotide or column missing) or there is an asterisk at *s*(*k*) or *a*(*k*) (nucleotide or column deleted), then *M*((*s*(*k*),*a*(*k*)) = *M*((*s*(*k*),*a*(*k*)) + 0.

Element *M*(*i*,*j*) indicated how many characters of type *i* occurred in position *j* for all family alignments. From matrix *M*, we calculated $y(i)=\underset{j}{\sum }m(i,j)$, $p\left(i\right)=y(i)/\underset{i}{\sum }y(i)$, where *p*(*i*) is the probability of occurrence of the *i*th nucleotide in all repeats from the family, and $N\left(j\right)=\underset{i}{\sum }m(i,j)$ is the number of nucleotides in the *j*th position of the alignment. Then, matrix *w*(*i*,*j*) was calculated as:(2)$$w\left(i,j\right)=\frac{\left(m\left(i,j\right)-N\left(j\right)p\left(i\right)\right)}{\sqrt{N\left(j\right)p(i)(1-p\left(i\right))}}$$

The elements of matrix *w*(*i*,*j*) characterized the degree of deviation of the observed nucleotide frequencies in various alignment positions from the random ones.

Then, for each alignment position *j*, we determined the value of $\chi (j)=\sum_{i}w{\left(i,j\right)}^{2}$ and calculated $X(j)=\sqrt{2\chi (j)}-\sqrt{2n-1}$, where *n* = 3 is the number of degrees of freedom. *X*(*j*) has an approximately normal distribution; the larger *X*(*j*) is, the greater the difference between base distribution in column *j* and random distribution. Figure 2 shows *X*(*j*) values for the first repeat family. The results indicated that there were highly conserved positions *j* with a large *X*. The graphs for all repeat families are shown in the Appendix A as fig2_<n>.jpg, where *n* is the family number from 1 to 20.

We obtained consensuses for the repeat families based on frequency matrix *MAT*, which was used to calculate matrix *M* (Section 4.1.3). The *j*-th symbol of the consensus was considered to be equal to the nucleotide present in more than half of positions *j* in all aligned sequences of the family; otherwise, the minus sign was used. An example of the consensus built with this algorithm is shown in Figure 3. All consensuses are presented in the Appendix A (freq_consensus.txt).

We also generated consensus sequences for each family by calculating multiple alignments of repeats from each generated family using the Weblogo 2.8.2 software [37]. For this, in each repeat sequence, we removed symbol *s*(*k*) if the opposite *a*(*k*) was a deletion (*k* = 1–600). In the Weblogo alignment, the height of each symbol was proportional to the frequency of its occurrence; the total height of all characters at a particular position was defined as the difference between maximum possible entropy calculated considering the equally probable occurrence of one of the four nucleotides at a specific position and the entropy observed for a given distribution of characters. The maximum possible entropy for the nucleotide alphabet was 2 bits. The Weblogo consensus for family 1 is shown in Figure 4, and the consensuses for all 20 families in the *png* format are shown in the Appendix A (fig4_<n>.jpg), where *n* is the family number from 1 to 20).

### 2.3. Intersection of the Found Dispersed Repeats with the Annotated Repeats in the C. merolae Genome

The data on the *C. merolae* genome annotation were extracted from http://plants.ensembl.org (accessed on16 April 2024) [38]. The annotation was completed using the Ensembl Genomes repeat feature; in the genomic sequence, the repeats are masked by symbols “*n*”. The number of repeats found by each method is given in Table 3. The total number of repeats previously found in the *C. merolae* genome is 29,211 (4,692,444 bp or 28.36% of the genome), including low-complexity regions identified by the DUST algorithm and tandem repeats identified by TRF, which we did not analyze in this study. As shown in Table 3, RepeatMasker has found very few dispersed repeats despite the use of different libraries, whereas RED demonstrated the best performance, being able to detect more than half of all repeats identified in the *C. merolae* genome. However, RED does not construct repeat consensuses, which is its significant disadvantage. Repeats annotated by different methods can overlap and even coincide.

We excluded the repeats of classes ‘dust’, ‘trf’, ‘Simple_repeat’, ‘Other/Simple’ and analyzed the intersection of the remaining 20,320 repeats designated as annotated dispersed repeats (ADRs, which, in total, constitute 4,648,259 bp or 28.09% of the *C. merolae* genome) with the repeats identified in this study.

The obvious difference between ADRs and the repeats found here was the repeat length. The minimum, maximum, and average lengths of the ADRs were 15, 19,220, and 260.5 bp. The ADR length distribution presented in Figure 5 (the right tail of the distribution corresponding to the longest 5% is not shown) indicates that the methods previously used to search for dispersed repeats in the *C. merolae* genome mostly recognize short repeats.

The intersection of the ADRs with the repeats found in this study was considered if the size of the overlapping region was more than 50% of the repeat length; one repeat could intersect with several ADRs and vice versa. It was observed that 14,421 (about 42%) of the repeats detected here overlapped with ADRs, indicating that more than half of the repeats were first identified in this study. At the same time, 16,103 ADRs overlapped with the repeats found here, indicating that we did not detect only 4217 ADRs. The reason for missing these repeats could be that in our algorithm, we used local PWM alignment of 600 bases chosen because it allowed for the detection of the largest number of dispersed repeats in the *C. merolae* genome. At the same time, many short repeats (<220 bases) were not detected, as indicated by the statistics of the repeat length distribution (Figure 1).

If we considered only ADRs over 100 bp, then set *T* of ADRs that we did not detect included only 1934 ADRs. It can be suggested that our method did not recognize sequences from set *T* because they have low copy numbers in the *C. merolae* genome, and since the families with less than 300 elements were not considered in our method, they were skipped. To test this hypothesis, we checked each of the 1934 ADRs from set *T* for the copy number in the *C. merolae* genome using BLASTN with default parameters. It turned out that the number of copies per genome for any set *T* sequence did not exceed 20, providing the reason for missing these dispersed repeats by the IP method.

We also examined the intersection of ADR classes with the repeat families constructed in this study. Since the functional significance of some ADRs is known, we looked for correlation between these ADRs and the 20 families identified with the IP method by considering all intersections of ADRs with the found repeat families. We calculated matrix *V* = {*v*(*i*,*j*)}, where *i* = 1, 2, …, 11 corresponds to the ADR class in Ensembl (DNA, DNA/En-Spm, DNA/hAT, LINE, LTR, LTR/Copia, LTR/Gypsy, MobileElement, Other, Repeatdetector, and rRNA), *j* = 1, 2, …, 20 is the number of the repeat family created here, and *v*(*i*,*j*) is the number of intersections between the ADR of class *i* and family *j*. Based on matrix *V*, we then calculated matrix *v*′ as:(3)$$v\prime \left(i,j\right)=\frac{\left(v\left(i,j\right)-np\left(i,j\right)\right)}{\sqrt{np\left(i,j\right)\left(1-p\left(i,j\right)\right)}}$$
where $x(i)=\sum_{j}v(i,j),$
$p(j)=\sum_{i}v(i,j)$, *p*(*i*,*j*) = *x*(*i*)*y*(*j*)/*n*, and $n=\sum_{i}\sum_{j}v(i,j)$. *v*′(*i*,*j*) was approximately normally distributed and showed the degree of correlation between the ADR classes and the found repeats; large *v*′*(i,j)* values indicated families enriched in ADRs. Matrix *V*′ is shown in Table 4. It should be noted that families 11, 12, and 13 identified here coincided with the ADRs of LTR/Copia and LTR/Gypsy classes.

## 3. Discussion

In this study, we showed that 72.64% of the *C. merolae* genome could be assigned to 20 families of dispersed repeats, although previously, only 28.09% of this genome has been considered to be occupied by repeats. To find divergent dispersed repeats in the *C. merolae* genome, we applied IP [30], which is a de novo method that does not require prior information about the repeat structure or use any databases of already known repeats. The IP method can detect highly divergent repeats with *x* up to 1.5 [30], whereas all previous methods can recognize repeats with *x* up to 1.0. The effectiveness of the IP method in finding weakly similar repeats that have accumulated a large number of mutations is due to the fact that instead of direct calculation of sequence alignment to determine similarity, this method constructs a PWM, which is an optimal image of multiple sequence alignment included in the family. The resulting PWMs function as templates to search for family members using dynamic programming and considering the correlation of neighboring bases [39]. Thus, the IP method can be applied to find repeats with a large number of indels, which are not recognized by k-mer-based methods; furthermore, the calculation of PWMs allows for building sequence consensuses for individual repeat families. These features distinguish the IP method from the other approaches, since PWMs can be applied to find repeats in different genomes using standard tools such as BLASTN or HMMER.

However, our method has certain limitations. First, the number of repeats in the family should be at least 300. Such a restriction is necessary because *N_max_* (Section 4.1.4) also includes randomly selected sequences and it is important to reduce their number. The average number of randomly included sequences is 122 (*σ* ≈ 12); for a family with 300 members, the deviation from a randomly expected family size is >10 *σ*, but the proportion of FPs is 122/300, which is about 30%. To reduce the FP rate, we searched for repeats by introducing our own threshold level Z_0_ selected independently for each found family. Therefore, the 72.64% of repeated elements in the genome calculated here is a minimum estimate.

The threshold *N_max_* = 300 is equal to the number of sequences that constitute *Mt_max_* (Section 4.1.4) and is the same for all repeat families found in this work. Only the number of observed families depends on *N_max_*, whereas their composition and the number of family members do not. If *N_max_* is reduced to 200, nothing will change in families 1 to 20; however, the number of the detected families will be greater than 20, in which case the contribution of FPs to *Mt_max_* for the extra families could be very significant and it will be impossible to correctly create *Mt_max_*. Therefore, it is difficult to accurately identify families with less than 300 repeats using our method, which is a limitation.

Second, the IP method works better if the repeat length is >220 bases. We tested this parameter using artificial sequences *S_a_*(*i*) (*i* = 1, …, 9) of 4 × 10^6^ DNA bases, into which a number (*N_a_*) of artificial dispersed repeats with length *L_a_* were inserted (*N_a_* was up to 500 and *L_a_* was from 50 to 600 bases); the repeats were created from a single mother sequence by introducing 0.5 *L_a_* random base substitutions and indels (~1 indel of a random size of 1–5 bases at random positions [every 50 bases in average]), which corresponded to *x* = 1.0. The results shown in Figure 6 indicate that when the IP method was applied to search for dispersed repeats in sequences *S_a_*(*i*) using a PWM with the number of columns equal to 600, it could detect dispersed repeats with length *L_a_* > 200 bases. However, if the number of random mutations was increased to 0.65 *L_a_* (*x* = 1.3), the method could find repeats longer than 300 bases.

Overall, the IP method could detect repeats with a length up 600 bases because in this work we used local alignment (Section 4.1.2) with *K*_0_ = −1 (*K*_0_ is the average of the PWM cell for all columns and rows, considering base frequencies [40]). The *K*_0_ = −1 value is optimal, since it allows for finding longer repeats at *x* > 1.0. It has been previously shown that with *K*_0_ < −1, it is possible to find dispersed repeats of < 200 bases [41]; however, in this case, the IP method could not identify long, highly divergent repeats for which *x* > 1.0 because it recognized them as multiple statistically insignificant short repeats. Since in this work the aim was to find repeats with *x* > 1.0, we chose *K*_0_ = −1; however to detect short dispersed repeats, previously developed algorithms such as RED [22] should be used.

Thus, the IP method could find dispersed repeats of >200 bases for *x* = 1.0 and of >300 bases for *x* = 1.3, which explains the results in Figure 1, showing that the lengths of the repeats detected in the *C. merolae* genome are over 300 DNA bases; these data correspond to the limitations of the IP method revealed using artificial sequences *S_a_*(*i*). Figure 5 indicates that the IP method omits most of the relatively short dispersed repeats in the genome of *C. merolae*, suggesting that repeated sequences could occupy a larger portion of its genome than the 72% we report here.

It is certainly possible to use PWMs with a column number ≤ 100; however, in such a case, it is more difficult to find indels in each repeat compared to the generated PWM. In this work, the number of columns in a PWM was chosen to be 600, which was optimal as it allowed for finding the largest number of repeats in the *C. merolae* genome; the use of other numbers of columns in PWMs would result in fewer identified repeat families and a consequential decrease in the calculated portion of the genome covered by repeated sequences.

The third limitation is related to the fact that the IP method searches for highly divergent repeats (1.0 ≤ *x* ≤ 1.5). The method first uses an iterative procedure to determine matrix *Mt_max_* (Section 4.1.4) containing 16 rows that correspond to base pairs, which means that in search for dispersed repeats, not only sequence similarity but also the correlation of neighboring bases are taken into account. The calculation of *Mt_max_* starts with a random matrix, which is used to search for small local maxima that may include dispersed repeats from different families with extremely insignificant similarity but with comparable correlations of neighboring bases. As a result of the iterative procedure, the matrix can simultaneously select two different repeat families and then identify these families when searching for dispersed repeats in direct and inverted forms (Section 4.2). In an illustrative experiment, we generated two families, each containing completely identical repeats of 600 DNA bases, which differed by the number of substitutions per nucleotide (*x*) from those of the other family (Figure 7). The graph shows that the two families can be identified by the IP method if there is no significant similarity between them (x > 1.7).

Such inclusion of two different families in one *Mt_max_* (Section 4.1.4) is possible, since *Mt_max_* has 16 lines, and each family can occupy, for example, 4 lines. In this case, matrix *Mt_max_* may still be non-random, which means that the localization of already known repeats in the *C. merolae* genome (Table 4) is not entirely accurate and that dissimilar sequences may be included in the same family found by the IP method.

The results shown in Table 4 indicate that LTR/Copia repeats are present in families 11, 13, and 15 and LTR/Gypsy repeats are present in families 11–13. It is most likely that the IP method placed the repeats of the two classes into the same family because of the existing similarity between the repeat sequences. It can also be seen that some other identified families are enriched in various classes of already known dispersed repeats (Table 4). We believe that it is possible to further improve the operative parameters of the IP method to isolate different families from matrix *Mt_max_*.

We examined the LTR/Copia and LTR/Gypsy repeats in more detail. Representatives of these classes belong to retrotransposons, which have long terminal repeats and include two genes, *gag* and *pol*, but may also contain other coding sequences. The gag protein is similar to the nucleocapsid protein of retroviruses, and pol is a multifunctional protein with protease, reverse transcriptase, ribonuclease, and integrase activities. The structural difference between LTR/Copia and LTR/Gypsy is the position of the integrase domain, which in the representatives of LTR/Copia is located upstream of reverse transcriptase and in those of LTR/Gypsy—after ribonuclease [9]. To compare LTR/Copia and LTR/Gypsy repeats present in the *C. merolae* genome, we generated two sets of sequences: *Q*_1_ containing annotated LTR/Copia-LTR/Gypsy repeats of *C. merolae* from the website (http://plants.ensembl.org (accessed on 16 April 2024); 2842 in total) and *Q*_2_ containing annotated LTR/Copia-LTR/Gypsy repeats overlapping with our families 11–13 (2430 in total). Scanning *Q*_2_ sequences against *Q*_1_ sequences using BLASTN revealed that out of 2430 *Q*_2_ repeats, 409 had significant similarities with those of the other class (LTR/Copia with LTR/Gypsy and LTR/Gypsy with LTR/Copia); examples of such similarities are provided in Appendix A (similarity_Gypsy_Copia.txt). The presence of similarities with extremely small E-values between repeats of different classes (*x* significantly less than 1.7; Figure 7) explains why they were placed by the IP method into one class of dispersed repeats.

The genome of *C. merolae* mostly consists of coding sequences, which include repeats as motifs. Our results on repeat consensus sequences indicate that members of the same family can significantly differ in the nucleotide composition; at the same time, we also detected conserved “islands” of a short length. We assume that conserved positions are binding sites for various proteins, probably histones or some other proteins. Similar results have been obtained after the alignment of highly smeared repeats in coding sequences of a bacterial genome [30].

The nucleosome repeat length (NRL) is a chromatin property important for its biological functions. The average NRL is about 150 bases [42], whereas the size of the repeats detected here is about 522 bases; therefore, it can be suggested that the repeats may include several nucleosome formation sites. Histones are constantly modified [43] and reassembled onto the DNA template at specific loci [44], which may correspond to the conserved islands within the repeat families. Changes in nucleosome positioning across the genome can result in rapid switching of the genetic activity of the cell [45,46], which is observed, for example, during cell differentiation [47,48]. Thus, dispersed repeats may be involved in gene regulation through participation in the dynamics of nucleosome formation.

Recent studies show that the bacterial genome, referred to as the “nucleoid”, has a well-defined substructure and dynamic behaviors [49]. Our previous analysis of bacterial genomes using the IP method has revealed dispersed repeats of several hundred bases long, which occupy more than half of the genome in many bacterial species [30], implying that these repeats could be involved in the nucleoid formation. A similar hypothesis can be applied to the results of the current study, suggesting that at least some of the identified dispersed repeats may play a role in the formation of nucleosomes and structural organization of the eukaryotic genome.

We are currently upgrading the IP method to use it for the annotation of longer eukaryotic genomes. Based on the results obtained, it can be expected that the proportions of dispersed repeats in the genomes detected by bioinformatics methods can be significantly increased.

## 4. Materials and Methods

### 4.1. Algorithm for Generating PWMs Specific for Dispersed Repeats of the C. merolae Genome

To search for PWMs of a family of dispersed repeats, we used the IP method [30]. A brief description of the method and its consecutive steps is presented in Section 4.1.1, Section 4.1.2, Section 4.1.3 and Section 4.1.4.

#### 4.1.1. Creating Random PWMs

We created set *Q* of 50 random matrices *M*(16,*L*), where *L* is the number of columns in the matrix. Each matrix *M* from set *Q* was filled with random numbers from −10 to 10 and then transformed so that $R^{2}=\sum_{i}\sum_{j}m{\left(i,j\right)}^{2}$, $K=\sum_{i}\sum_{j}m(i,j)p_{1}(i)p_{2}(j)$, and *R*_0_ was always equal to *K*_0_. Here, *p*_1_(*i*) = *f*(*k*)*f*(*l*), (*f*(*k*) and *f*(*l*) are the probabilities of encountering bases *k* and *l*, respectively, in nucleotide sequence *Sw* [Section 4.1.2]), and *p*_2_(*j*) = 1/*L*. In this work, we used *L*_1_ = 600, $R_{0}^{2}=300L^{0.5}$, and *K*_0_ = −1.0 as in [30]. The transformed matrix was denoted as *Mt*. The procedure for transforming matrices *M* is described in detail in [40].

#### 4.1.2. Searching for PWM-like Sequences in the *C. merolae* Genome

Each *Mt* from set *Q* was used to search for similar sequences in the *C. merolae* genome. For this purpose, all chromosomes were merged into one sequence denoted as *S* with length *L_S_*. In sequence *S*, we isolated a window of 650 DNA bases denoted as *Sw*(*x*), where *x* is the coordinate of the first base in *S*, and calculated the local alignment between matrix *Mt* and sequence *Sw*(*x*); then, we determined similarity function *F_max_* as described earlier ([30], Section 4.3). Briefly, we first created window *Sw*(*x*) for *x* = 1, calculated *F*(*t*) = *F_max_*(*x*) (where *t* = int(*x*/10) + 1)), added 10 bases to *x*, calculated local alignment, and determined *F*(*t*) = *F_max_*(*x*) again. As a result, we obtained vector *F*(*t*) for *x* from 1 to *L_S_*-650 and determined the local maxima for *F*(*t*), i.e., those *t* for which *F*(*t* − *i*) ≤ *F*(*t*) ≥ *F*(*t* + i), where *i* varied from 1 to 65.

In order to evaluate the statistical significance of the found local maxima, we determined the average value $\bar{F}$ and standard deviation σ(*F*) by randomly shuffling sequence *S* to obtain sequence *S_rand_* and to determine vector *F_rand_*(*t*), $\bar{F}$, and σ(*F*). Then, we calculated vector *Z*(*t*) for sequence *S* as $Z(t)=(F(t)-\bar{F(t)})/\sigma (F(t))$ and selected only those local maxima that had *Z*(*t*) > *Z*_0_(1) (*Z*_0_(1) value was chosen to be 3.0). The number of local maxima for which *Z*(*t*) > *Z*_0_ was denoted as *N_z_*(1) and the coordinates of local maxima were *G*(*i*), *i* = 1, 2, …, *N_z_*(1).

#### 4.1.3. Creation of a New PWM Based on the Local Maxima

In the next step, we collected all local alignments of matrix *Mt* from set *Q* that corresponded to the local maxima for which *Z*(*t*) > *Z*_0_. Each alignment had two sequences: one of DNA bases denoted as *S*_1_ ([30], Section 4.4) and the other of columns in matrix *Mt* denoted as *S*_2_. Then, we filled frequency matrix *MAT*(16,600) as:*MAT*(*n*,*s*_2_(*i*)) = *MAT*(*n*,*s*_2_(*i*)) + 1(4)
for all *i* from 1 to *k*. Here, *k* is the length of the local alignment of sequences *S*_1_ and *S*_2_, *n* = *let*(*s*_1_(*i* − 1)) + 4(*let*(*s*_1_(*i*)) − 1), *let*(*a*) = 1, *let*(*t*) = 2, *let*(*c*) = 3, and *let*(*g*) = 4, which means that *n* varied from 1 to 16. If in the alignment *s*_1_(*i* − 1), *s*_1_(*i*), or *s*_2_(*i*) had negative values, i.e., contained a deletion, then Equation (4) was not satisfied and we moved to *i* = *i* + 1. The use of *s*_1_(*i* − 1) and *s*_1_(*i*) in dynamic programming ([30], Section 4.2) allows for the construction of a local alignment, taking into account the correlation of neighboring bases.

Matrix *MAT* was filled for all local maxima whose coordinates were recorded in *G*(*i*), and new matrix *M* was calculated as:(5)$$M\left(i,j\right)=\frac{MAT\left(i,j\right)-Np(i,j)}{\sqrt{\left(N)p(i,j)(1-p(i,j)\right)}}$$
where *p*(*i*,*j*) = *x*(*i*)*y*(*j*)/*N*^2^, $x(i)=\sum_{j=1}^{600}MAT(i,j)$, $y(j)=\sum_{i=1}^{16}MAT(i,j)$, and $N=\sum_{i=1}^{16}\sum_{j=1}^{600}MAT(i,j)$. Then, matrix *M* was converted into matrix *Mt*, as described in Section 4.1.1.

#### 4.1.4. Iterative Procedure to Search for PWMs and Create Repeat Families

Matrix *M_t_* calculated in Section 4.1.3 was again used to search for local maxima in the *C. merolae* genome, as described in Section 4.1.2. However, in the second iteration, we used *Z*_0_(2) = 5.0 and, as a result, obtained a new number of local maxima *N_z_*(2) and their coordinates *G*(*i*), where *i* = 1, 2, …, *N_z_*(2). For the found local alignments, we again calculated matrices *MAT* and *Mt*. Then, the cycle was repeated 20 times using *Z*_0_ = 5.0, and series *N_z_*(*i*) (*i* = 1, 2, …, 20) were obtained. Finally, we chose the iteration for which *i* > 8 and *N_z_*(*i*) had the maximum value; the number of this iteration was denoted as *i_max_*. The threshold of *i* > 8 was chosen to exclude sharp fluctuations in *N_z_*(*i*).

We performed similar calculations for all matrices from set *Q*, obtained *i_max_*, *N_z_*(*i_max_*), and the coordinates of local maxima *G*(*i*) (*i* = 1, 2, …, *N_z_*(*i_max_*)) for each matrix, and chose the one with the largest *N_z_*(*i_max_*) denoted as *N_max_*; the corresponding matrix was denoted as *Mt_max_* and repeat coordinates as *G_max_*(*i*) (*i* = 1, 2, …, *N_max_*). *Mt_max_* was considered as the matrix of the family of dispersed repeats and *N_max_* as the number of dispersed repeats in the family.

After the formation of the first family of dispersed repeats characterized by *Mt_max_* and *N_max_*, we marked all bases in the local maxima of sequence *S*, which allowed for excluding all repeats included in the family from further consideration. Then, we repeated all the calculations described above to find other repeat families. During this search, *Z*(*t*) (Section 4.1.2) was set as zero, if the alignment included at least one labeled base from sequence *S*, which allowed us to construct subsequent families of dispersed repeats that did not intersect with each other. The formation of repeat families continued until *N_max_* > 300.

Calculations described in Section 4.1.1, Section 4.1.2, Section 4.1.3 and Section 4.1.4 were performed using the online resource http://victoria.biengi.ac.ru/shddr/auth/login (accessed on 16 April 2024), which allows for the identification of repeat families and calculation of their *Mt_max_* matrices for entered DNA sequences of ≥1 million bases. The calculation time for 16 million bases was about 72 h.

### 4.2. Search for Repeats in Both DNA Strands

To reduce the calculation time needed to construct repeat families in Section 4.1., we considered one DNA strand, which indicates that direct and inverted repeats should form two rather than one family. In order to correctly identify repeat families by considering both DNA strands, we created matrix $Mt_{\max}^{inver}$ by rotating *Mt_max_* 180 degrees along the columns and exchanging rows between complementary bases, and then searched for sequences similar to both *Mt_max_* and $Mt_{\max}^{inver}$, as described in Section 4.1.2. As a result, for each family of repeats, we obtained local maxima *Z*(*t*) and *Z*(*t*)*^inver^* as well as their coordinates *G* and *G^inver^*.

To exclude the emergence of the same genomic sequence in different families, we intersected coordinates *G* of all found families and identified the overlapping repeat sequences, i.e., those that had a common fragment of more than 50 bases. Among the overlapping repeats, we selected only one with the largest *Z*(*t*) and excluded all the others. The same procedure was performed for coordinates *G^inver^* of all inverted repeats. The programs used in Section 4.1.4 and Section 4.2 are shown in Appendix A. This Appendix A also contains instructions for using them.

## Figures and Tables

**Figure 1 ijms-25-04441-f001:**
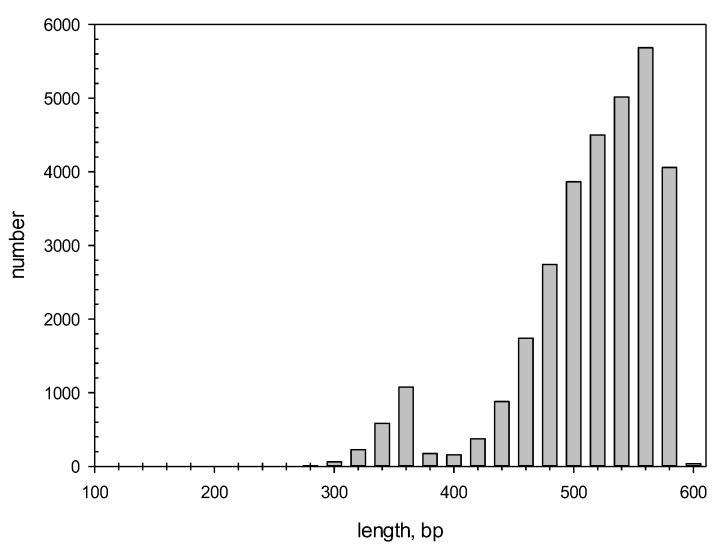
Length distribution for the repeats detected by the IP method.

**Figure 2 ijms-25-04441-f002:**
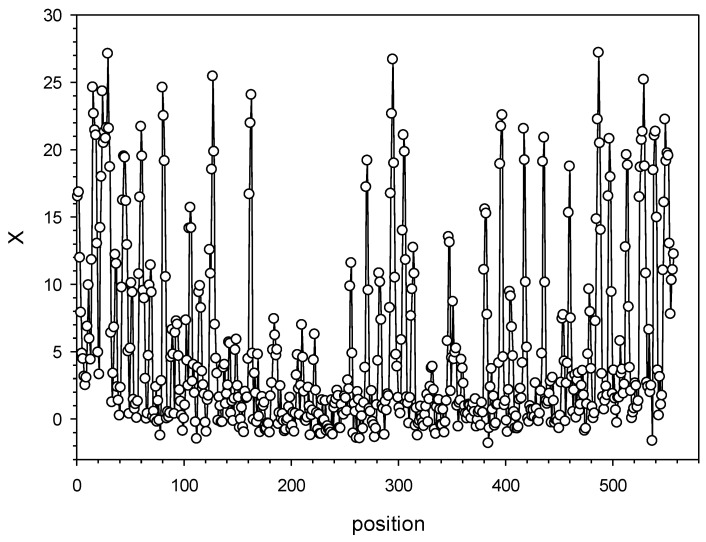
Dependence of *X* on profile position *j* for the first repeat family.

**Figure 3 ijms-25-04441-f003:**
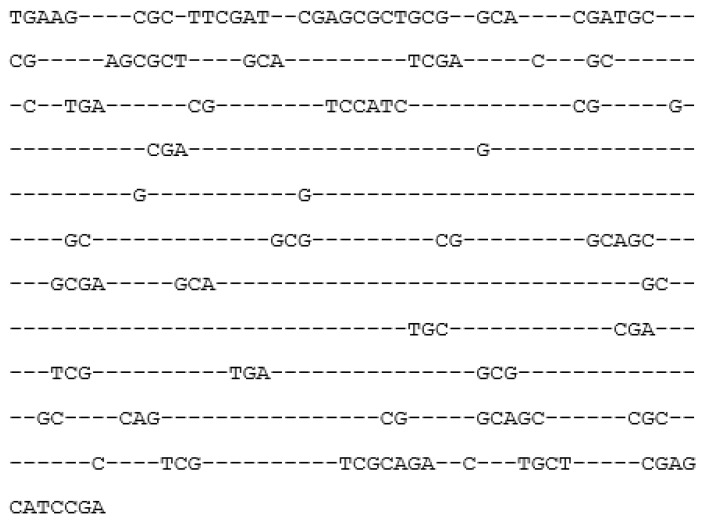
Symbolic consensus for the first repeat family.

**Figure 4 ijms-25-04441-f004:**
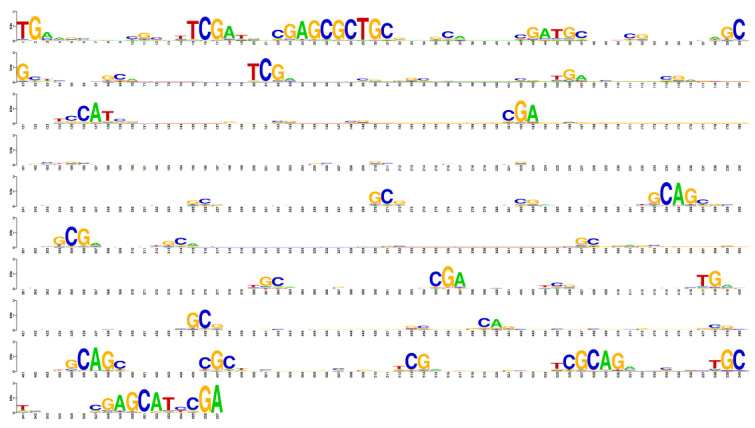
Weblogo-built consensus for the first repeat family.

**Figure 5 ijms-25-04441-f005:**
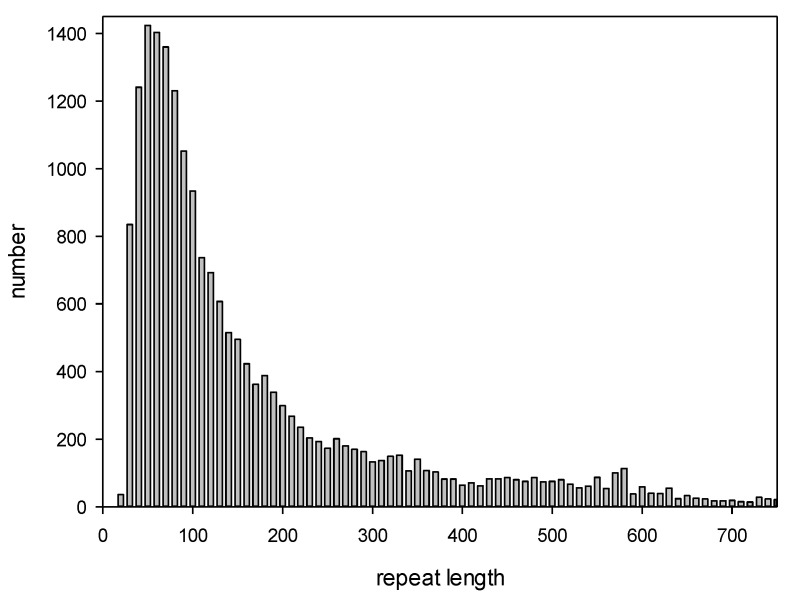
Histogram of ADR lengths.

**Figure 6 ijms-25-04441-f006:**
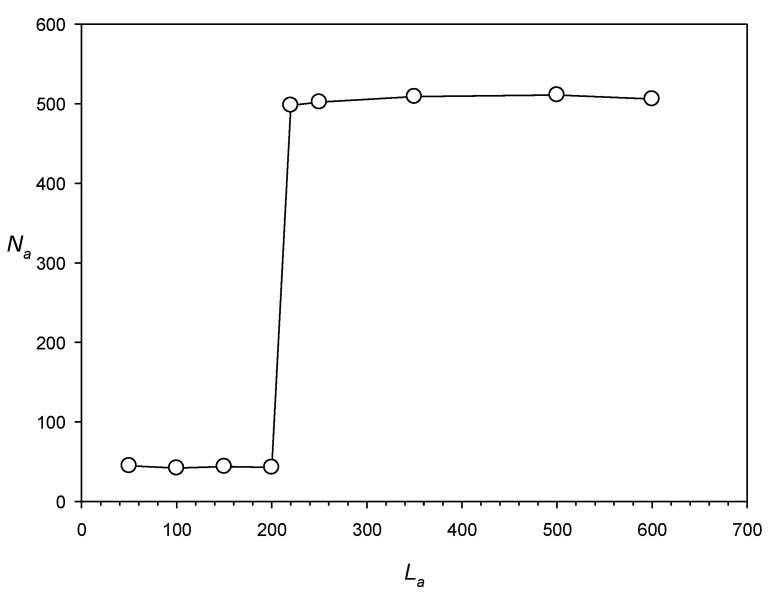
Dependence of the number of dispersed repeats *N_a_* found in sequences *S_a_*(*i*) on the length of dispersed repeats *L_a_*. Details of constructing *S_a_*(*i*) are given in the text.

**Figure 7 ijms-25-04441-f007:**
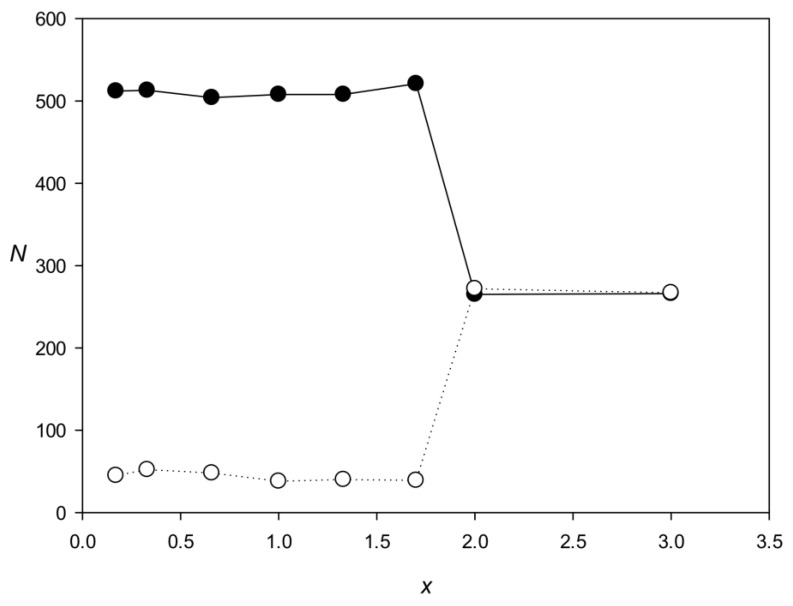
Search for two families of identical dispersed repeats (600 bases in length) differing in the *x* value. Black and white circles indicate the two families. The number of members in each family *N* was 250.

**Table 1 ijms-25-04441-t001:** Number of found repeats in each family of the *C. merolae* genome (*C. merolae*) and randomly shuffled genome sequence (Rand).

Family	*C. merolae*	Rand	FDR
1	4465	48	1.06%
2	1996	19	0.94%
3	1677	45	2.61%
4	1874	29	1.52%
5	1462	20	1.35%
6	3284	49	1.47%
7	1313	20	1.50%
8	1712	19	1.10%
9	1156	32	2.69%
10	3084	43	1.38%
11	2202	17	0.77%
12	1117	33	2.87%
13	2039	25	1.21%
14	643	18	2.72%
15	1687	14	0.82%
16	719	19	2.57%
17	1196	27	2.21%
18	915	30	3.17%
19	612	13	2.08%
20	785	32	3.92%
Total	33,938	552	1.60%

**Table 2 ijms-25-04441-t002:** Number of found repeats in each chromosome on the forward (+) and reverse (−) strands and in total (+ and −).

Chromosome	Chromosome Size, bp	+	−	+ and −
1	422,616	347	461	808
2	457,013	357	489	846
3	481,791	392	518	910
4	513,455	407	560	967
5	528,682	495	553	1048
6	536,163	554	540	1094
7	584,452	596	629	1225
8	739,753	781	763	1544
9	810,151	934	824	1758
10	839,707	960	872	1832
11	852,849	961	742	1703
12	859,119	1023	738	1761
13	866,983	1021	734	1755
14	852,727	1001	915	1916
15	902,900	1017	934	1951
16	908,485	989	927	1916
17	1,232,258	1266	1229	2495
18	1,253,087	1379	1185	2564
19	1,282,939	1414	1178	2592
20	1,621,617	1836	1413	3249
Chloroplast	149,987	4	0	4
Mitochondria	32,211	0	0	0
Total	16,728,945	17,734	16,204	33,938

**Table 3 ijms-25-04441-t003:** Numbers of repeats found by repeat feature pipelines of Ensembl Genomes.

**Program**	**Number**
DUST	5266
TRF	3335
RepeatMasker, database Redat	1773
RepeatMasker, database Repbase	2102
RED	16,735
Total	29,211

**Table 4 ijms-25-04441-t004:** Correlation between ADR classes and repeat families created with the IP method.

Repeat Families	ADR Classes										
	DNA	DNA/ En-Spm	DNA/ hAT	LINE	LTR	LTR/ Copia	LTR/ Gypsy	Mobile Element	Other	Repeat Detector	rRNA
1	2.47	0.49	1.81	−0.79	4.20	−9.22	−6.88	0.70	0.28	3.83	4.09
2	−0.23	5.97	0.72	1.42	1.55	−5.44	−4.50	−0.46	−0.57	2.26	−0.07
3	−0.19	0.54	−1.92	1.90	1.41	−5.57	−5.40	−0.38	−0.47	3.18	−0.86
4	−0.24	0.12	4.31	−0.53	3.51	−5.77	−1.43	−0.48	2.84	0.99	−1.07
5	−0.21	2.37	−1.10	−0.46	2.73	−5.25	−2.24	−0.41	−0.51	1.48	2.32
6	−0.29	−1.51	3.65	0.89	0.71	−6.92	−7.22	−0.58	−0.71	4.03	1.78
7	−0.16	−1.50	−1.65	−0.37	−2.51	−3.63	−7.20	2.70	−0.40	4.11	−0.74
8	−0.19	−1.19	2.73	−0.43	−0.45	−3.74	−5.22	−0.39	−0.47	2.77	0.29
9	−0.15	−0.70	−1.54	−0.35	−0.78	−3.75	−4.11	−0.31	−0.38	2.69	−0.69
10	−0.29	2.71	0.25	0.91	4.27	−7.14	−8.84	1.16	−0.71	4.10	0.27
11	−0.31	−1.37	−3.07	−0.69	−6.99	23.65	15.08	−0.61	−0.75	−9.76	−1.37
12	−0.19	−1.77	−1.94	−0.43	−2.05	−5.65	14.47	−0.39	−0.48	−3.47	−0.87
13	−0.34	−1.44	−3.07	−0.75	−7.04	24.94	25.55	−0.67	−0.82	−14.13	−1.51
14	−0.13	−1.22	−1.34	−0.30	−0.85	−3.77	−6.11	−0.27	−0.33	3.46	−0.60
15	−0.25	−1.80	−1.24	−0.55	−4.30	9.89	−1.86	−0.49	−0.60	−0.63	−1.10
16	−0.12	0.80	−1.17	−0.26	−0.15	−3.12	−4.27	−0.23	−0.29	2.37	−0.53
17	−0.19	−1.09	3.01	−0.41	2.94	−4.18	−1.64	−0.37	1.75	0.96	−0.83
18	−0.15	0.02	1.70	−0.35	2.01	−4.14	−3.39	2.93	2.27	1.71	−0.69
19	−0.12	0.72	−0.38	−0.27	2.21	−3.00	−3.42	−0.24	−0.30	1.61	−0.54
20	−0.14	−0.44	−1.37	2.97	4.74	−2.32	−1.51	−0.27	2.66	0.38	−0.61

Values greater than 3.0 and smaller than −3.0 are highlighted green and red, respectively.

## Data Availability

All data supporting reported results can be found at Appendix A.

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
