# Peer review of "Study of Dispersed Repeats in the Cyanidioschyzon merolae Genome"

_ijms, 2024, doi:10.3390/ijms25084441_

Round 1

Reviewer 1 Report

Comments and Suggestions for Authors

In this manuscript, Rudenko et al. applied the previously proposed iterative process (IP) method for searching highly dispersed repeats in the eukaryotic genome, and identified repeats by constructing position weight matrices (PWMs) and dynamic programming. The authors conducted experiments on the genome of C. merolae and found 20 repeat families, which occupied 72% of the genome. Although this article provides some valuable insights into dispersed repeat identification, there are still some parameters of the experimental design that have low clarity. Additionally, there are also a couple of minor mistakes that need to be fixed.

My major comments are as follows:

1. The method proposed in this manuscript only detects dispersed repeats with lengths ranging from 100 to 600 bp, while shorter or longer repeat sequences, which are commonly found in ADR, cannot be detected. Please explain the limitations of this method in addressing this issue. In such a scenario, the conclusions drawn from the experiment results may not accurately represent the true range of repeat lengths and their proportions in the genome. For instance, in the Abstract (page 1), it is stated, "The repeats varied in length from 108 to 600 bp (522.54 bp in average) and occupied more than 72% of the C. merolae genome." Please clarify this point to avoid potential reader misunderstandings.

2. Figure 1, page 5. Why are there almost no repeats in the 100-300 bp in this figure, which contradicts the annotations in ADRs? What is the reason for this inconsistency?

3. The authors point out that due to the high complexity of the IP method, research has been conducted on the short genome of C. merolae. Would it be feasible to consider studying model organisms with small genome sizes (e.g., Drosophila)? If not, please provide the reasons.

4. Section 2.2, page 4. For each family, “new families were considered if the number of repeats in a family exceeded 300.” Would it be possible to conduct comparative analyses with families meeting different threshold values, thus providing a more rational approach to threshold setting?

5. Table 2, page 4. As a complement to the information provided in Table 2, I suggested to include circos plots. These plots would display the distribution of dispersed repeats within the chromosomes, thereby yielding more valuable insights.

6. Table 4, page 10. As described in the manuscript, “Table 4 shows that families 11, 13, and 15 mostly contain repeats of the LTR/Copia type, whereas families 11–13 comprise those of the LTR/Gypsy type” (paragraph 4, page 11). It is puzzling that family 13 exhibits a high correlation with both LTR/Copia and LTR/Gypsy types. Therefore, I recommended to explore the structural characteristics of repeat sequences within family 13 through more detailed comparisons.

Some minor comments:

1. Paragraph 2, page 9. “The minimum, maximum, and average lengths of the ADRs were 15, 19.220, and 260.45 bp” implies that the maximum length is less than the average length. Please verify the specific numerical values.

2. The declaration of “Q” in the first line of the last paragraph on page 9 conflicts with the statement of “Q” in Section 4.1.1 (page 12). Please consider using different character identifiers to avoid confusion.

Comments on the Quality of English Language

Minor editing of English language required.

Reviewer 2 Report

Comments and Suggestions for Authors

The present paper by Rudenko and Korotkov deals with a very important issue on genomics. Repetitive DNA annotation is challenging, and therefore, all contributions to the subject are welcome.

Here are some comments on the paper:

Although I understand the authors point, I found that assign 90% of repetitive content to "fish" is not accurate. I suggest that authors use something like, some fish species can reach up to 90% (...).

In the same direction, authors describe the abundance of TEs as if there is a general pattern for all eukaryotes (since nothing is said about which organisms are they taking about). For example "The second TE class, which is much less numerous than class I". Not for every organism this sentence is true.

In fact, authors fall into this generalization at the beginning "The dominant type of dispersed repeats in the eukaryotic genome is transposable elements (TEs)". Not for every organism this sentence is true.

On the other hand, the authors explained extremely well the scenario for detecting and annotating repetitive sequences, specially TEs.

About results, on the discussion authors explain that "Table 4 shows that families 11, 13, and 15 mostly contain repeats of

the LTR/Copia type, whereas families 11–13 comprise those of the LTR/Gypsy type." My question is, 11 and 13 are gypsy or copia? authors do not discuss this result. Maybe could be that they are dealing with a LTR inserted in another LTR and one is copia and the other in gypsy. But nothing is said about this ambiguous result, which is depicted in the discussion.

On the discussion, authors state that "The very large proportion of dispersed repeats discovered here suggests that the identified repeat families may play a role in the processes of genetic regulation." However, this is no further discussed. In my opinion, this statement is not within the aim of this paper. What is more, the concept is no further discussed and no further data is provided to support the hypothesis. At the very least, authors should provide more evidence from other papers and further discuss this topic. Otherwise I recommend the authors to eliminate the sentence.

Reviewer 3 Report

Comments and Suggestions for Authors

This work presents an interesting investigation into the identification of dispersed repeats in the eukaryotic microalga C. merolae, utilizing a method previously proposed by the authors. It the work could potentially expand the application of their dispersed repeats identification tool, although several questions and comments arise:

1. The lengths of repeats detected by their method exhibit significant discrepancies from those of the annotated ones. The authors should provide justification for the PWM matrix window size of 600 bases. I wonder how the output looks like if the L decreases to a smaller one, for instance, to 400 or 200 bases?

2. It is desirable to see the comparative analysis or, at the very least, a discussion comparing this study to the authors' previous work in E. coli, exploring the method's performance and applicability across different organisms.

3. Section 2.1 is not the results, and should be moved to the Introduction.

4. The method description of consensus sequence calculation in section 2.3 should be moved to Materials and Methods.

Comments on the Quality of English Language

The quality of English is acceptable.

Round 2

Reviewer 3 Report

Comments and Suggestions for Authors

My questions are addressed. I have no further ones.

Comments on the Quality of English Language

The quality of English language is acceptable.